# An Overview of Twenty-Five Years of Augmented Reality in Education

**Juan Garzón** (ID)

Faculty of Engineering, Universidad Católica de Oriente, Sector 3, cra. 46 No. 40B 50, Rionegro, Antioquia, Colombia; fgarzon@uco.edu.co

**Abstract:** Augmented reality (AR) enables an interactive experience with the real world where real-world objects are enhanced with computer-generated perceptual information. Twenty-five years have passed since the first AR application designed exclusively to be used in educational settings. Since then, this technology has been successfully implemented to enrich educational contexts providing learning gains, motivation, enjoyment, autonomy, among other benefits. This study provides an overview of AR technology in education from its origins to the present. Consequently, based on the analysis of its evolution, the study defines three generations of AR in education. Moreover, the study identifies some major challenges from previous AR applications and, finally, it poses some insights to address these challenges to enhance the benefits of AR for education.

**Keywords:** artificial intelligence; augmented reality; education; smartglasses; systematic review; WebAR

## 1. Introduction

Augmented reality (AR) enables an interactive experience with the real world where objects in the real world are enhanced by computer-generated perceptual information. This technology has positively influenced different fields, such as industry, entertainment, medicine, tourism, among others. Nevertheless, experts assure that this is only the beginning and that the future of AR will bring better, cheaper, and more accessible applications [1].

This technology has also left a positive mark on education. Twenty-five years have passed since the development of the first AR application designed exclusively to be used in educational settings. Since then, AR applications have been successfully implemented at different levels of education, different fields of education, and different educational environments providing multiple benefits for learners [2]. However, there are still some pending issues that must be addressed to obtain the best of this technology to enrich education. Furthermore, it is important to note that as AR hardware evolves, AR applications will evolve, presenting new affordances and challenges for the AR research area.

This study provides an overview of AR technology in education from its origins to the present day. It identifies its history, status, and trends in the educational context and shows how educational AR applications have undergone different phases since the first application in 1995. Consequently, based on the analysis of its evolution, the study proposes three generations of AR technology in education. Subsequently, the study identifies some pending issues from previous applications and finally, it highlights some insights to solve such issues to enhance the benefits of AR for education.

## 2. Overview of AR in Education

According to records in the scientific and academic literature databases, the first AR system designed exclusively to be implemented in educational settings was a tool for teaching three-dimensional anatomy. This AR tool superimposed and registered bone structures on real anatomical counterparts of a human subject to teach anatomy using a head-mounted display. The system was developed at the University of North Carolina and

introduced by its creators in the first International Conference on Computer Vision, Virtual Reality, and Robotics in Medicine held in Nice, France, in 1995 [3].

The period from 1995 to 2009 was characterized by applications based on head-mounted displays and heads-up displays. The Web of Science (WoS) reports 80 AR studies in education published in that period, which focused on AR applications to complement the learning processes in the fields of Health, Engineering, and Natural Sciences. It is also important to note that most of these applications were intended for undergraduate students and that only a few educational institutions could afford these applications given their high costs [4]. Thus, we could describe the first 15 years of AR in education as a transition period because of some notable limitations that yielded a small number of AR applications. Additionally, the scope of these applications was narrow, considering that apparently only three fields of education and one level of education were benefiting from them.

A new wave of AR arrived with the emergence of game engines, Software Development Kits (SDKs), and libraries to develop AR applications [5]. Former Microsoft Developer Consultant Lester Madden documented the rise of mobile AR as a positive consequence of the emergence of Wikitude, Layar, and Junaio ushering in a new era for AR applications [6]. Madden highlighted two main reasons for this situation. First, these SDKs make it easy to build content with little or no programming skills. Second, it is not necessary to have expensive and complex systems to develop or use the application, as everyday devices such as smartphones provide all the necessary hardware to bring AR experiences to life. This may be considered the outbreak of mobile AR, posing 2010 as a milestone for AR applications. This panorama was completed in 2011 with the release of Vuforia, the current most popular SDKs for developing educational AR applications [7]. Google Glass and Pokémon Go gave another significant boost to AR in the middle of the past decade. Google Glass was released in 2013 for qualified developers as a brand of smartglasses developed by Google X. A year later, the company released the version for the public seeking to empower people's lives with information before their eyes. On the other hand, Pokémon Go, a mobile game developed by Niantic, was released in 2016. Only nineteen days later, it reached 50 million users, becoming the most popular smartphone application ever, and consequently, one of the drivers of AR's popularity [8]. Consequently, the last decade of AR in education has flourished with many applications that have complemented learning processes in all fields of education and at all levels of education [9].

The future of AR in education also appears to be encouraging. The internationally recognized series of Horizon Reports claims that new technologies such as AR will lead to a redesign of learning and teaching [10]. Horizon Reports argues that the increased use of AR has enabled mobile learning to become more active and collaborative, creating limitless learning experiences. However, this report states that it is not only the technology that needs to be engaged with, but also the educational outcomes that it is seeking to achieve. In this regard, as stated by the study by Garzón, Kinshuk, Baldiris, Gutiérrez, and Pavón [11], future AR applications should consider not only technical characteristics but also an appropriate pedagogical approach that potentiates the affordances of AR for education. This panorama, united to the development of emerging scenarios for AR such as smartglasses, AR-based Web (WebAR), and the integration of Artificial Intelligence (AI) seems to pose unthinkable opportunities to enrich educational contexts.

### 2.1. Status and Trends of AR in Education

To identify the status of AR in education, we conducted a search in the WoS Website on 13 July 2020 using the search term "augmented reality in education". One document type parameter was modified to be adjusted to the purpose of the search. Document type included only articles, proceeding papers, reviews, and book chapters, and excluded meeting abstracts, editorial material, corrections, and book reviews. The rest of the parameters were left as default. This search resulted in 2698 studies, including 1317 proceeding papers, 1857 articles, 100 reviews, and 56 book chapters. Figure 1 sorts the studies by year of publication, where the first study of AR in education in the WoS was published in 1996. It

is important to mention that the first AR system was presented in 1995 in a conference and is not available in the WoS.

**Figure 1.** Number of studies of AR in education per year in the WoS.

Figure 1 shows a steady increase in the number of studies of AR in education. Furthermore, data analysis highlights two important facts. First, a logistic model indicates a latency period from 1996 to 2009, and then, 2010 marks an inflection point toward an exponential growth. This sets 2010 as a milestone for educational AR applications, which is in line with the outbreak of mobile AR posed by Madden [6]. Second, the most noticeable increases occurred from 2014 to 2015 and from 2016 to 2017. This coincides with two specific events, the release of the public version of Google Glass in 2014 and the released of Pokémon Go in 2016. These two events brought AR to prominence technologies, attracting many developers worldwide to create AR applications for education.

To complete the analysis of the status of AR in education, we sorted the studies by country, university, and language. Accordingly, data indicate that the Unites States (347), Spain (273), Taiwan (114), China (113), and Germany (110) are the five most representative countries. Similarly, the State University System of Florida (31), University of La Laguna (23), Polytechnic University of Valencia (23), Harvard University (21), and University of Sevilla (21) are the most representative universities. According to the languages, data indicate that English (1898), Spanish (94), Portuguese (20), German (6), and Russian (5) are the most used languages to communicate AR in education-related studies.

To identify the trends regarding the level of education, fields of education, and delivery technology, we based our work on the studies by Garzón et al. [9]; Sirakaya and Sirakaya [12]; and Akçayir and Akçayir [13]. The study by Garzón et al. analyzed 61 studies published between 2012 and 2018, the study by Sirakaya and Sirakaya analyzed 105 studies published between 2011 and 2016, and the study by Akçayir and Akçayir analyzed 68 studies published between 2007 and 2015. Although the studies were based on different databases, WoS and Scopus (Garzón et al.), ERIC, EBSCOhost, and ScienceDirect (Sirakaya and Sirakaya), and Social Science Citation Index (Akçayir and Akçayir), they found similar results. The studies found that AR applications have been successfully implemented to teach most fields of education. However, based on the definition of the

International Standard Classification of Education [14], the most popular broad field is Natural sciences, mathematics, and statistics, followed by the broad fields of Arts and Humanities and Health. On the other hand, the studies found that AR applications have been implemented at most levels of education. Based on the definition of the International Standard Classification of Education [14], the most popular levels of education are Bachelor, followed by Secondary education and Primary education. Finally, the results indicate that the most popular delivery technology is mobile devices, followed by computers and head-mounted displays.

### 2.2. Three Generations of AR in Education

Based on the analysis of its evolution, we pose three generations of AR applications in education (see Figure 2). The first generation covers the period from 1995 to 2009 and could be described as hardware-based AR, as the delivery technology was the protagonist of the AR experience. The second generation covers the period from 2010 to 2019 and could be described as application-based AR, as the AR experience focused on AR applications rather than AR hardware. Finally, the third generation runs from 2020 onward and seems to be characterized by dedicated AR devices such as smartglasses and Web-based AR.

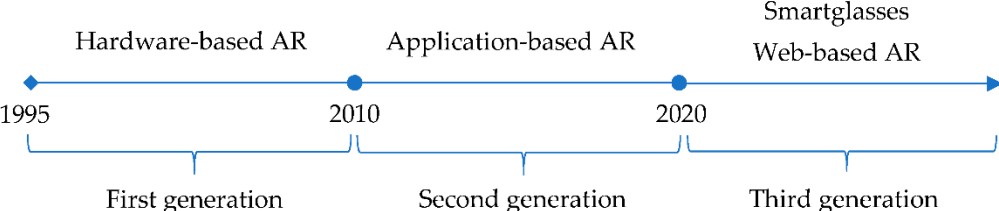

**Figure 2.** Three generations of AR in education.

#### 2.2.1. First Generation of AR in Education (1GARE)

This generation was characterized by expensive and complex AR systems based on devices such as head-mounted displays, heads-up displays, and handheld displays. All these systems were intended to teach subjects related to health, natural sciences, or engineering, and focused on bachelor students as target groups. Applications of the 1GARE presented two main limitations, namely, high costs and usability. The high cost of 1GARE applications was related to expensive hardware devices, the need for specialized maintenance of hardware devices, and the need for special programming skills for content development [15]. Consequently, the acquisition of AR educational applications was unfeasible for most educational institutions, which did not make the effort to afford them, considering the multiple available educational technology tools that provided similar benefits. Another major factor affecting 1GARE applications was usability. Usability refers to the quality of a user's experience when interacting with the application, and it is related to the ease of use of the application [16]. These applications were complex systems that caused the overexertion of the users. Consequently, cognitive overload to master the applications hindered the learning processes. Because of high costs and low usability, 1GARE applications were unpopular in educational environments, thus affecting their dissemination [17].

Billinghurst [18] published one of the first analyses of AR in education. This analysis described AR as an especially valuable technology for education as it can support seamless interaction between real and virtual environments. However, the limited number of available user studies did not allow it to identify the real impact of AR on education. Similarly, Dunleavy, Dede, and Mitchell [19] described the evolution, affordances and limitations of AR in education. This is considered the first analysis of user studies and concludes that the novelty effect becomes a motivating factor for students to learn to use AR applications. However, the study predicts the possibility that the novelty engagement

will fade as students become accustomed to this method of learning, and consequently they warned of an uncertain future for AR in education.

2.2.2. Second Generation of AR in Education (2GARE)

This generation began with a significant increase in the number of AR applications for education [20]. This outbreak obeys in part to the solution to the two main limitations of the 1GARE. First, the fact that AR applications could be deployed in mobile devices meant that more people could access AR, eliminating the need to purchase expensive devices. Additionally, the emergence of game engines, SDKs, and libraries to develop AR applications enabled easier methods to develop AR applications, saving time and money. Moreover, as mobile devices became everyday use technology, it somehow guaranteed that users could easily use AR applications. This generation was benefited from the release of the public version of the Google Glass in 2014 and Pokémon Go in 2016. These two novel applications raised the interest of a great number of developers and users of AR technology, positioning it among the most relevant alternatives in educational technology.

Yuen, Yaoyuneyong, and Johnson [21] published a study that may be considered the boundary between 1GARE and 2GARE. This study stated that AR has substantial potential implications and numerous benefits for teaching and learning. Based on the analysis of previous research, they proposed five directions for educational applications: AR books, AR gaming, discovery-based learning, object modeling, and skills training. The study posed, accurately, that such directions would guide the future of AR in education. Later that year, Carmigniani et al. [22] published the first systematic literature review of AR in education. The review analyzed 25 studies published between 2002 and 2010 to describe the most important applications of AR in education. This study described mobile AR as a successful megatrend that will potentially play a critical role in education. The study highlighted that AR was socially accepted thanks to its integration into mobile devices, expanding people's skills and senses beyond the sense of sight. Two years later, Wu, Lee, Chang, and Liang [23] published the most cited study of AR in education. This study describes the status, opportunities, and challenges of AR in educational settings and proposes possible solutions to limitations of 1GARE applications. Akçayir and Akçayir [13] analyzed 68 studies to identify the advantages and challenges associated with AR for education. The study highlights learning gains and motivation as the main advantages of AR in education, while the most important challenge is the inclusion of pedagogical strategies to accompany AR interventions. Finally, Garzón and Acevedo [2] conducted a meta-analysis of 64 studies published between 2010 and 2018 to measure the impact of AR on students' learning gains. They found an effect size of 0.64 to conclude that AR has a medium impact on education according to Cohen's classification [24].

2.2.3. Third Generation of AR in Education (3GARE)

Although AR has become one of the most interesting technologies in the digital industry, trends seem to indicate that it has not reached its full potential. In this sense, the future of AR appears to evolve beyond head-mounted displays and smartphones, with three main scenarios that have emerged recently, namely, smartglasses, WebAR, and AI.

The first scenario characterizes by stand-alone headsets such as HoloLens, Oculus Rift, or the upcoming iGlass, which are part of a new generation of smartglasses. According to ABI Research, shipments of smartglasses worldwide will rise from 225 thousand units in 2017 to around 32.7 million in 2022 [25]. This poses a new frame for the future of AR applications, that will potentially influence industry, medicine, tourism, education, among others, providing important advantages over other forms of technology.

The second scenario appears as a solution for the reluctance of some people to experience AR via smartphone applications. To experience AR on smartphones, users must install a dedicated application. However, users often remove it after a few uses and other potential users do not take the trouble to download it at all. Hence, in 2017, WebAR appears to enable smartphone users to discover AR technology via the Web, eliminating

the installation process [26]. Thus, while still less efficient compared to application-based AR, WebAR has the potential to revolutionize the Web and take AR technology to the next level.

Finally, the third scenario brings together AR and AI to create solutions to different problems of everyday life [27]. This union bridges the physical and digital worlds, moving the boundaries of the digital world beyond screens and into the multisensory 3D world. AI fuels AR, enabling the shift to more realistic and engaging experiences and a more powerful customization of applications. Consequently, the integration of AR with AI opens a new direction for developers and researchers to explore and experiment, which will potentially create a renewed paradigm for AR.

## 3. Pending Issues of AR in Education

Despite the multiple proven benefits of using AR in educational settings, there are some pending issues that need to be addressed to enhance the impact of AR on education. Based on the previous literature, this study identifies four major issues that should guide future research directions, namely, accessibility, usability, dissemination, and pedagogical approaches.

### 3.1. Accessibility

The first issue is related to accessibility, which mainly affects people with some type of disability. Accessibility refers to the design of applications in a way that can be used by all people regardless of their special needs [28]. In education, the term special needs refers to students who have some type of disability, including blindness or vision impairment, deaf or hard of hearing, intellectual disabilities, and physical disabilities. According to the World Health Organization (WHO), more than one billion people worldwide live with some disability, a number that is expected to double by 2050 [29]. Despite some efforts and the proven benefits of AR for special needs education, most AR applications for education still lack accessibility characteristics [9]. A study by the Georgia Tech's Wireless Engineering Rehabilitation Research Center stated that 92% of people with disabilities use wireless devices such as smartphones or tablets [30]. Thus, it can be inferred that mobile technologies play a central role in providing autonomy to students with any type of disability. In this sense, the incorporation of accessibility features in AR applications has the potential to bring the numerous advantages of this technology to enhance special needs education.

### 3.2. Usability

The second issue refers to usability, which is related to the ease of use of AR applications. In fact, different studies have indicated that the most reported challenge of AR in education refers to the complexity of using AR systems [9,13,31,32]. Although the usability of 2GARE applications improved notably compared to 1GARE applications, some situations affect the quality of users' experiences when interacting with AR applications. As pointed by Akçayir and Akçayir [13], AR involves multiple senses and requires simultaneous tasks from students, which may overload their attention, affecting the usability of AR systems. Therefore, it is important to consider design strategies that favor the usability of AR applications to ensure that they can be easily implemented in any educational context. Additionally, it is important to identify how new deployment technologies can improve AR systems' usability, to help overcome this challenge of AR in education.

### 3.3. Dissemination

The third issue is related to dissemination, which derives from aspects such as cross-platform deployment and the need for download and installation. Most AR applications are designed to be implemented on a specific platform and lack cross-platform support, which affects their dissemination. Consequently, to reach more users, developers of AR applications must go through repeated development cycles to accommodate multiple

platforms, increasing production time and costs [1]. On the other hand, application-based AR requires a process of downloading and installation that takes additional time and effort. This causes users to often delete the application after a few uses to save space on their devices and other potential users do not bother to download it at all. Therefore, to improve dissemination, it is necessary to reduce the number of obstacles the end user faces to access the AR experience, which can be achieved by eliminating the need to download or update AR applications.

### 3.4. Pedagogical Approaches

Finally, the fourth issue is related to the lack of pedagogical approaches when integrating AR applications into learning activities. This issue has been identified by different studies, signaling that in most cases teachers use AR applications without considering pedagogical aspects, reducing the effectiveness of the interventions [33,34]. In this sense, Garzón et al. [9] claim that educational applications based on AR must transcend technological aspects, as the technology by itself does not ensure success in the learning process. Instead, it is a combination of the technological affordances and the pedagogical approaches that powers AR's impact on education. To complement this point, different studies indicate that the lack of formal pedagogical approaches when applying AR to learning activities tends to confuse and frustrate students [35,36]. Therefore, it is necessary to identify which pedagogical approaches are the most appropriate for each educational setting, to encourage stakeholders to consider technology together with pedagogical strategies to guarantee the best of AR for education.

## 4. Insights to Enhance the Benefits of AR for Education

The affordances of the 3GARE suggest a promising future for educational AR applications. In this section, we indicate how emerging AR technologies such as smartglasses and WebAR, along with AI, can provide solutions to overcome some of the pending issues of AR in education.

### 4.1. Insights to Improve the Accessibility of AR Applications

While progress on assistive technologies has been made, there are still barriers that prevent users with special needs from having efficient AR experiences. Moreover, it is important to recognize that special needs education includes a broad set of needs, each of which requires a different solution [37]. In this regard, new AR technologies have the potential to turn AR into a digital sixth sense that enables people to master skills in ways that are not possible with other available technologies.

From our point of view, smartglasses are a key component to improve the accessibility of AR systems. According to the WoS, the first smartglasses-based application was published in 2017. To July 2020, 11 studies have reported smartglasses-based applications to be used in educational contexts. We highlight that 7 out of the 11 studies were focused on special needs education, which is an encouraging figure in terms of social inclusion. Smartglasses are embedded with several components, such as camera, compass, calculator, thermometer, accelerometer, speaker, microphones, and navigation system. In addition, smartglasses are powered with AI, which allows them to provide multiple solutions to different types of needs.

One of the groups that potentially benefited the most from smartglasses is blind or visually impaired people. Computer vision technology has played a significant role in assisting this group to carry out daily activities. However, research on accessibility for this group has focused on specific tasks, mainly text-entry. The inclusion of screen reading software on smartphones has popularized AR applications amongst this group; nevertheless, users still face several problems opening and using basic functionalities when interacting with touch interfaces. Smartglasses uses computer vision techniques (powered by AI) to extract information from the context and convert it into audio files in almost any required language. Additionally, it is important to mention that all smartglasses' features

can be activated by voice, eliminating the need to touch or identify the context. This way, by the simple point of a finger, blind or visually impaired people can read almost any text in front of them and can access the multiple functionalities of the device [38]. Additionally, smartglasses allow document scanning to turn the document into speech and image reader to understand a picture by describing scenes, colors, and objects in the image. These features provide blind and visually impaired people enough autonomy to carry out daily activities, without much dependency on other people. This autonomy has created the possibility for this group of people to attend educational contexts, almost under the same conditions as non-visually impaired students [39].

Another group that benefits from smartglasses are individuals with hearing impairments, including the deaf or hard of hearing. Most educational applications for this group focus on adding subtitles to videos or audio files. On the other hand, the integration of this group in educational contexts relies on an interpreter who translates what the teacher says. However, students with hearing impairments must keep turning away from the interpreter to take notes, which means wasting valuable seconds of communication in the classroom. Smartglasses can take audible information and convert it to visual information or add captions to it. Furthermore, these devices allow students to receive live sign language interpretation superimposed on the classroom environment [40]. The content of the teacher's speech can be delivered to the user through the smartglasses by using speech-to-text technologies or by displaying a sign language interpreter. In using speech-to-text technologies, the speaker's speech is converted to text and the text is then displayed inside the smartglasses [41]. With the smartglasses on, users do not need to look away, which means that they can continue to look at the teachers, while understanding what they are saying.

Among the group of intellectual disabilities, people with autism spectrum disorder (ASD) are apparently the group that benefited the most from smartglasses. People with ASD may be socially, communicatively, and emotionally impaired, with symptoms that include stereotypic and tantrum behaviors, self-harm, isolation, passivity, and withdrawal. This situation frequently yields learning difficulties and lack of positive interactions with their surrounding environment [42]. In this regard, different studies point out that smartglasses hold the key to improving social and communications skills of people with ASD [27,42]. The studies show acceptable tolerability of this assistive technology, providing benefits regarding sensory, cognitive, and attentionally challenges. These systems combine emotion-based AI and smartglasses, which keep users engaged in the social world by encouraging direct interaction, unlike smartphone-based applications that immerse users in a screen. Additionally, users remain hands-free, so they can use their hands to participate in non-verbal social communication and undertake educational activities [27]. Consequently, studies report improvements in social interactions, through improvements in non-verbal communication, social engagement, and eye contact. Additionally, the studies report reductions in ASD symptoms including irritability, lethargy, stereotypic behavior, hyperactivity, and inappropriate speech.

Finally, people with physical disabilities that cause motor impairment such as quadriplegia, cerebral palsy, lost or damaged hands, muscular dystrophy, arthritis, among others, can also benefit from smartglasses. People who suffer from any of these disabilities may need to use devices such as head wands, mouth sticks, among others. However, the smartglasses' features provide multiple benefits that allow them to participate in educational scenarios, without much dependency on other people or invasive assistive technologies. All smartglasses' features can be activated by voice, which means that this group of people can take notes, take pictures, record scenes, and audios, etc., without having to move their hands or any other part of the body except their mouth.

### 4.2. Insights to Improve the Usability of AR Applications

Ko, Chang, and Ji [43] published one of the most comprehensive studies regarding AR applications' usability. The study proposed five groups of usability principles for

AR applications that have guided the development of some 2GARE applications, namely, user-information, user-cognitive, user-support, user-interaction, and user-usage. User-information includes indications on the hierarchy to present the contents, the use of proper language, and multimodality to present the contents. User-cognitive is related to cognitive aspects required to minimize memory and cognitive overloads. User-support includes providing instructions for use, a help section, a friendly and adaptable interface, and support for potential errors. User-interaction is related to the responsiveness of the application, the ability to provide feedback to users, and the need to minimize users' effort when using the application. Finally, user-usage seeks to guarantee availability, context-based interface design, and fluency in navigation. However, despite the proven benefits, different studies have indicated that including these principles to AR applications demands too much work, time, and money. Consequently, developers do not always strive to design usable AR applications, making it the most reported challenge for AR in education.

As an alternative, we pose that WebAr-based applications have the potential to improve the usability of AR systems. The main advantage of WebAR over other AR applications, as for usability, is that people are familiar with Web-based applications and, consequently, using WebAR-based applications happens to be naturally easy [44]. Moreover, as WebAr does not require users to download any specific application, all they need is to do is visit a website in their devices' browser to enjoy the AR experience. This simplicity translates into great usability, promoting better engagement, which will potentially enhance the popularity of WebAR in educational settings.

However, whatever the format of the AR experience, interaction with AR requires high motor skills, spatial cognition, and attention control. Therefore, developers must consider special characteristics when designing AR systems. Based on the usability components defined by the ISO 9241-11 guidance on usability [45] and the principles established by Ko et al. [43], we highlight three insights to address AR applications' usability. The first component refers to the application's goals. As for this component, it is important to consider both the educational goals and the technical goals. The pedagogical goals must consider the accuracy of the educational content, a low cognitive load, the motivating elements, and the feedback. On the other hand, the technical goals must look for transparent integration between the real world and the virtual world. As for the educational content, it must be presented in multiple formats as each user has their own needs and preferences. Additionally, these contents should be skippable and non-playable to guarantee efficiency and productivity. Lastly, as AR applications also seek to provide autonomy, it is important that AR applications provide help and documentation. The second component refers to the context of use of the application. The context includes the level of education, the field of the education, and the learning environment. One efficient way to address the needs of the application regarding the context of its use is the inclusion of an instructional design model. These models bridge the pedagogical and technical worlds to produce pedagogically accurate and technically efficient AR applications. These models include phases of analysis, design, development, implementation, and evaluation that guide the development process of educational applications. Additionally, it is important to consider specific pedagogical approaches when designing the application. As stated by Garzón et al. [11], the integration of a specific pedagogical approach to a given educational context will improve the impact of AR in educational interventions. Finally, the third component refers to usability measures. This is important to verify the correct usability of the AR application before its implementation in an educational context in terms of efficiency, effectiveness, and user satisfaction. Based on the usability guidelines of ISO 9241-11 and the study by Ko et al. [43], we establish six key elements to measure AR applications' usability. These elements are related to (1) match between the real and virtual worlds, (2) efficiency of use, (3) consistency between virtual elements and educational purposes, (4) aesthetic and minimalist design, (5) help and documentation, and (6) user satisfaction.

### 4.3. Insights to Improve the Dissemination of AR Applications

Hardware-based AR and application-based AR have issues that have hindered AR from having broader dissemination in educational settings. On the one hand, hardware-based AR implementation is known to be expensive and lacks flexibility. On the other hand, application-based AR involves an additional upfront download and installation process, which impacts cross-platform deployment and increase production costs and development time. Alternatively, as it has been noted, to access WebAR, all users require is a mobile device and a browser, which has the potential to encourage more users to experience AR. Although still in its early stage, WebAR promises pervasiveness to AR systems, as it supports almost all Android and iOS devices worldwide (Android 6.0 and above, and iOS 11 and above) [26].

Another affordance that WebAR brings to education is the possibility of creating educational content through authoring tools [46]. Authoring tools allow developers to create educational applications by using preprogrammed elements, eliminating the need for specific programming experience [47]. By using authoring tools, inexperienced developers can create educational content in the format of learning objects, that can be reused in other educational contexts. Therefore, the integration of authoring tools into the field of WebAR will empower inexperienced teachers and developers to create educational content, also promoting the dissemination of AR.

In another context, recent years have brought together AR and AI as complementary technologies, making AR experiences more realistic and engaging. These technologies work seamlessly together to create personalized experiences that adapt to each students' needs and preferences [48]. AI helps designers use data about students' preferences to tailor educational content to each student's situation. Additionally, AI helps designers learn from every student interaction, thereby enhancing the delivery of the pedagogical content [49].

Deep learning is another important benefit that AI brings to AR. Deep learning involves machine learning algorithms to model high-level abstractions in data using computational architectures that support multiple and iterative nonlinear transformations of data [50]. Deep learning achieves unprecedented levels of accuracy that can be applied to computer vision, speech recognition, natural language processing, and audio recognition. In summary, it provides smaller, faster, and more accurate models that are replacing traditional approaches underpinning AR experiences [39]. Hence, AI-powered AR systems are more attractive to a broader spectrum of students, which will translate into more AR users.

### 4.4. Insights for Integrating Pedagogical Approaches into AR Experiences

The affordances of the 3GARE are not only related to new deployment technologies, but also to pedagogical strategies to enhance the impact of AR on education. Different studies indicate that the success of AR educational experiences depends not only on the technical quality of the applications but also on the pedagogical strategies to implement them. The study by Videnovik, Trajkovik, Kiønig, and Vold [51] proposes an analysis of the quality of the learning experiences using AR educational games. The study concludes that many games are introduced into education without a methodological approach, just because students like to play and that, on the other hand, there are many educational games that are not interesting to play. Hence, the study proposes a set of methodological guidelines based on design thinking to provide a link between pedagogical approaches and entertainment to achieve a successful integration of games into education. A similar study by Lester and Hofmann [52] poses AR as a disruptive technology that needs to be considered in terms of its pedagogical implications as well as its effectiveness as a learning tool. The study focuses on vocational education and training since, as mentioned by the authors, the emphasis on the use of AR in vocational contexts is mainly focused on technical implementation issues. The authors propose some pedagogical observations on the use of AR in vocational practices to achieve long-term benefits for vocational students. They

conclude that AR tends to create changes in training methods that go beyond learning, when it is linked to appropriate pedagogical strategies.

Garzón et al. [11] conducted the first meta-analysis of the 3GARE to identify, in the light of the learning theories, how the pedagogical approaches affect the impact of AR on education. The study analyzed 46 empirical studies published between 2010 and 2019 to conclude that AR interventions that included pedagogical approaches showed better results than AR interventions that did not do it. Additionally, the study presents recommendations for stakeholders on how to design AR interventions to obtain the best of AR technology for education. The study poses that the unique characteristics of AR have the potential to support and enhance a variety of pedagogical approaches, including situated learning, collaborative learning, inquiry-based learning, and project-based learning. Furthermore, the study notes that results in individual studies may vary depending on a wide range of factors, such as the pedagogical approach, the learning environment, the learner type, and the domain subject.

The study concluded that, as a rule, collaborative learning is the approach that most benefits AR interventions. However, situated learning is the approach that better impacts subjects related to engineering and it is the most popular pedagogical approach in AR interventions. In general, interventions conducted in informal settings performed better compared to interventions conducted in formal settings. However, formal settings favor AR interventions that include subjects related to Social sciences. Similarly, the most effective pedagogical approach in informal settings is project-based learning, while collaborative learning obtained the best results in formal settings. Regarding the level of education, the most benefited target group seems to be Bachelor-or-equivalent-level students, and there is no pedagogical approach that benefits a particular target group the most. The results of this study provide a broad overview of the inclusion of pedagogical approaches to educational AR experiences, and therefore can be used as the bases for the design of AR interventions in the 3GARE.

Finally, as a framework to integrate the pedagogical approaches in AR interventions, we highlight the use of instructional models such as Biggs' 3P model [53]. This model proposes that the learning experience consists of three stages, namely, presage, process, and product. The presage stage includes the analysis of students' prior knowledge, abilities, preferences, and expectations. Additionally, it includes curriculum, teaching method, classroom climate, and assessment from the teaching context. Process stage refers to how the elements of the presage stage influence the task processing determining the learning styles. Lastly, the product stage involves the nature of the learning outcomes as a product of the interaction of the elements in the process stage.

## 5. Conclusions

This study presents an overview of twenty-five years of AR in educational settings. Based on the evolution of this technology, we defined three generations of AR in education. The first generation (1995–2009) can be defined as hardware-based AR as it focused on the delivery technology. The popularity of AR in education was limited due to some issues related to high costs and low usability. Consequently, educational AR applications covered only some topics related to Health, Engineering, and Natural Sciences and most of them were intended for undergraduate students.

The second generation (2010–2019) can be defined as application-based AR, as it focused on AR applications rather than AR hardware. The integration of AR on mobile devices greatly increased the popularity of AR in education as it reduced costs and increased usability. Consequently, AR applications reached each field of education and each level of education, positioning it as a novel technology for enhancing education. However, the existing literature indicates that despite notorious improvements in educational AR applications, some issues such as accessibility, usability, dissemination, and pedagogical accuracy prevent AR from having a better impact on education.

The third generation (2020 onwards) evolves through two different scenarios: smartglasses and WebAR. These scenarios are enriched by AI, with the promise of turning AR into a mature technology to complement each educational context. Smartglasses make part of a branch of wearable technologies that have the potential to impact our lives similarly to smartphones. However, smartglasses' characteristics bring additional benefits to special needs education. Smartglasses include a broad set of advantages over other technologies, such as activation and control by voice commands, non-invasive display devices, no-need for touchscreens, and others. These advantages empower users with different needs, providing them with autonomy, social integration, motivation, enjoyment, and the promise of a more inclusive and dignified future. For its part, WebAR provides pervasiveness to AR as it does not require users to download or acquire specific applications or hardware, which makes it the least demanding way of executing AR. Consequently, it is one of the most accessible and usable platforms to experience AR, which makes AR in the Web a new norm. WebAr enables developers to design and implement AR applications in an easier and less time-consuming way than traditional methods. In education, it yields that inexperienced stakeholders are empowered to develop Web-based AR applications by using authoring tools, providing more students with the benefits that AR brings to education.

There is a third scenario, AI, which although is not an AR technology per se, powers the applications of the 3GARE. These two technologies work together to bridge the physical and digital worlds, opening a new realm for stakeholders to explore and experience education at a higher level. AI models have become incredibly good at doing many of the things required to build immersive AR experiences. Therefore, upcoming AR applications are expected to be more realistic, user-centric, cheaper, accurate, and omnipresent thanks to the affordances provided by AI.

Finally, it is important to note that a successful educational AR application depends not only on the technical issues but also on the pedagogical characteristics of the context in which it is used. Each application must be specifically designed to satisfy the needs and preferences of users in each educational context. Thus, teachers and researchers must be aware that a specific application may be successful in a specific context but not be successful in another context. This study attempts to provide insights on how to use AR technology to enrich education and what can be expected from new developments to improve the experience of using AR technologies.

**Funding:** This research received no external funding.

**Institutional Review Board Statement:** Not applicable.

**Informed Consent Statement:** Not applicable.

**Data Availability Statement:** Not applicable.

**Conflicts of Interest:** The author declares no conflict of interest.

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
