# Peer review of "An Overview of Twenty-Five Years of Augmented Reality in Education"

_mti, doi:10.3390/mti5070037_

Round 1

Reviewer 1 Report

The manuscript contains a discussion of research on the topic of AR in Education.

The paper is well written and might be interesting to read for someone who is not familiar with the field. For an expert, the paper does not provide any new information apart from a discussion of trends and insights in the end.

The literature search in WoS presented in the beginning is very simple and only looks at the number of papers per year in a single (arguably, not the most representative for a rapidly developing field) database.

The analysis of literature reviews is presented, but a gap in the research is not identified. Therefore, the need for the present paper is not justified.

Extensive self-citation raises concerns over the additional value of this paper. The self-cited articles are of the same type - discussion papers.

Author Response

The response to reviewer 1 comments can be found in the attached document.

Reviewer 2 Report

In this study, 25 years of AR in education is reviewed, resulting in the description of 3 generations and 4 pending issues. It was a pleasure to read: a comprehensive and nicely written article. 

I only have some minor remarks and advise to accept the paper after these issues have been addressed adequately.

1) The title of section 4.2 is the same as 4.1 - I assume in 4.2 Accessability should be replaced by Usability

2) Section 4.2 is somewhat superficial compared to 4.1, 4.3 and 4.4. Could you maybe elaborate on models to improve usability (in general or specifically for AR-systems)? The lack of the need to download and install is not enough, especially since this is strongly related to the Dissemination issue as well.

3) In section 4.4, you may add models for instructional design to enhance the pedadogical underpinning of the design and development of AR in education, e.g. the Carpe Diem model as a process model or the 3P-model (or similar ones) for the learning content & activities.

Author Response

The response to reviewer 2 comments can be found in the attached document.

Reviewer 3 Report

I really appreciated the article. It's well written and it's relevant. It's a contribution for junior researchers and for countries where Augmented Reality is still not a reality such African countries. 

Author Response

Thank you very much for your positive comments on the paper.

Reviewer 4 Report

The paper provides a very good overview of AR in education. The organization of the paper presented some very good research points that is crucial to the continuous research of the field. The pending issues described in section 3 are dead on pointing out the current issues preventing meaningful progression of the field.  

However, section 4.1 and 4.2 have similar title with the two sections elaborating different insights.  In addition, following the general sentiment on AI/ML research trend, the author also included the use of AI/ML/DL technique as the final promising technology that will finally solved all the research problems on AR in education without offering any promising concrete research idea, path or existing research in the field.  

Author Response

The response to reviewer 4 comments can be found in the attached document.

Round 2

Reviewer 1 Report

No further comments.